# Lithium Niobate Piezoelectric Micromachined Ultrasonic Transducers for high data-rate intrabody communication

Flavius Pop [1✉], Bernard Herrera[1] & Matteo Rinaldi[1✉]

In recent years, there has been an increased interest in continuous monitoring of patients and their Implanted Medical Devices (IMDs) with different wireless technologies such as ultrasounds. This paper demonstrates a high data-rate intrabody communication link based on Lithium Niobate (LN) Piezoelectric Micromachined Ultrasonic Transducers (pMUTs). The properties of the LN allow to activate multiple flexural mode of vibration with only top electrodes. When operating in materials like the human tissue, these modes are merging and forming a large communication bandwidth. Such large bandwidth, up to 400 kHz, allows for a high-data rate communication link for IMDs. Here we demonstrate a full communication link in a tissue phantom with a fabricated LN pMUT array of 225 elements with an area of just 3 by 3 mm square, showing data-rates up to 800 kbits/s, starting from 3.5 cm and going up to 13.5 cm, which covers the vast majority of IMDs.

[1] SMART Center, Electrical and Computer Engineering Department, Northeastern University, Boston, MA, USA. ✉email: flavius.pop@ieee.org; m.rinaldi@northeastern.edu

The Internet of Medical Things (IoMT) has been the subject of wide research aimed at creating networks of medical devices that continuously acquire medical data from patients and make it available to both the user and healthcare providers[1]. Wearable devices such as smartwatches and wrist bands[2] are examples of commercially available devices that provide a diverse collection of health information (heart rate, exercise, sleep patterns, etc.) and can easily connect to the internet[3]. Implantable Medical Devices (IMDs)[4] are a second category of IoMT devices than can provide data that is more specific to certain organs and parts of the body given their closer proximity to the site of interest due to implantation. These devices also offer site-specific stimulation capabilities for treatment purposes. However, the transmission of this data from within the body to the external world is challenging. The goal of this work is to bridge the communication technology gap between the IMDs and the final beneficiaries of the data—the patients and the doctors. This is achieved by creating innovative technologies for intrabody communication links[5–7] that allow wireless communication between implanted and non-implanted devices, and at the same time keep the patient safe by complying with the Food and Drug Administration (FDA) regulations[8].

Ultrasonic technology has emerged as a viable candidate for implementing intrabody communication, given the wide variety of implantation depths IMDs require[9]. The reason being that ultrasonic waves are mechanical in nature and therefore propagate with lower attenuation in human tissue at the frequencies of interest, around 1 MHz. The first demonstration of intrabody communication utilizing ultrasound technology was performed with bulk piezoelectric transducers based on Lead Zirconate Titanate (PZT)[10]. Good performance in terms of data rate and penetration of the acoustic waves into the body was shown. However, PZT transducers contain lead which make these devices non-bio-compatible[11] and requires extra packaging for implanting as part of an IMD. Moreover, their bulky form factor is a further disadvantage in terms of invasiveness for the patients, which also introduces a high risk of infection and rejection from the body. Thus, recent research has been focused on Micro-machined Electro-Mechanical Systems (MEMS), such as Capacitive Micromachined Ultrasonic Transducers (cMUTs)[12] (working principle in Methods) and Piezoelectric Micromachined Ultrasonic Transducers (pMUTs)[13] (working principle in Methods) as shown in Fig. 1. For both devices, the key advantage is their miniaturization capability through standardized semiconductor micro-fabrication processes. The main advantage of pMUTs over cMUTs is that they don't require a DC bias to be actuated, therefore making them more appealing for CMOS integration (less complex design and lower power consumption). On the other hand, the key advantage of the pMUTs over bulk transducers is the biocompatibility of the main piezoelectric material they use, the Aluminum Nitride (AlN)[14] as opposed to the PZT[11] used in the bulk transducers. This allows for the implantation of pMUTs alongside IMDs without the need of additional packaging. A second advantage is the possibility to integrate the pMUT fabrication directly in CMOS foundries, lowering the final product cost and shortening the time to market. Nonetheless, the AlN has a smaller electro-mechanical coupling coefficient compared to other piezoelectric materials such as PZT, meaning that it generates less output pressure when applying the same AC voltage. Recent work has replaced the AlN with micromachined PZT showing better performance but losing the advantage of biocompatibility[15]. The latest promising option is to dope the AlN with Scandium (ScAlN)[16], which allows to increase the electromechanical coupling factor. Preliminary results have shown similar performances to that of PZT while maintaining the biocompatibility of the pMUT.

A key challenge for intrabody communication is to increase the data rate. In fact, operating at ultrasound frequencies such as 1 MHz avoids signal attenuation[17], but in return, it limits the data rate. This is opposed to what can be achieved with RF antennas operating at much higher frequency but having a dramatic increase of attenuation in tissue as shown in Fig. 2c. In water, the EM waves attenuate with a rate of 0.2 dB/cm while the US waves attenuate at a rate of 0.002 dB/cm; even though the US waves attenuate much less, it only introduces a difference of 3 dB at 15 cm, which is beyond the maximum range of our experiment. However, when considering the tissue phantom, the EM waves attenuate at a rate of 4 dB/cm while the US waves attenuate only at a rate of 0.7 dB/cm; this introduces a difference of 50 dB at 15 cm when operating in tissue phantom, which is closer to the human body in terms of attenuation and acoustic impedance, hence making the US waves a better fit for intrabody communication[18–20] such as in our application with the LN pMUTs. The main concept, to increase the data-rate, is to design devices with a large operation bandwidth. PMUTs based on AlN and ScAlN already present a wider bandwidth compared to bulk PZT transducers, cMUTs, or even PZT-based pMUTs[3,15,16].

In this work, we show a better alternative by designing arrays of pMUTs that include elements centered around different frequencies. These frequencies are closely spaced to cover a certain desired bandwidth. Once the pMUT array is implanted or submerged in a liquid medium, the different resonance frequencies will merge together, due to the damping effect, and achieve a large bandwidth. However, this poses challenges at the fabrication level to precisely tune the resonance frequency of each individual pMUT. In order to ultimately increase the communication bandwidth, the proposal of this work is to introduce a different piezoelectric material, the X-cut Lithium Niobate (LN), which allows to harness multiple resonance frequencies in one device which can therefore merge into a larger bandwidth when operating in a body-like medium. At the same time, by harnessing stronger piezoelectric coefficients[21] compared to AlN and ScAlN, the LN pMUTs are able to cover a broad range of implantation depths and result in higher data rates for the communication schemes.

## Results

**Experimental setup.** In this work, we are demonstrating an X-cut LN-based pMUTs. Implemented at the system level, this allows the creation of wide band and high-data-rate intrabody communication links with a high implantation depth range. In Fig. 1 we are showing the design of a single LN pMUT (a), the design of an entire array of pMUT (b), the Scanning Electron Microscope (SEM) image of a fabricated device (c), and ultimately an optical image of a fabricated array (d). For more details, the step-by-step fabrication process is described in Methods. In order to determine the remarkable properties of the LN pMUTs, we are characterizing these in air and in tissue-like media. The characterization in air consists of measuring the 3D membrane displacement of the pMUT's membrane with a Digital Holographic Microscope (DHM). The measurements match the resonant modes predicted by a Finite Element Simulation (FEM) as shown in Fig. 2a. More details are found in Methods. These results show that, with only two top electrode activation, we can excite multiple resonance modes around the main peak and that these are very distinctively separated for in-air operation. Following, the characterization in tissue-like media (de-ionized water, silicone oil, castor oil, or tissue phantom), consists of transmitting a high intensity and short duration signal with a reference hydrophone and then receiving this ultrasonic signal at a certain distance with an LN pMUT array. More details about the characterization setup are

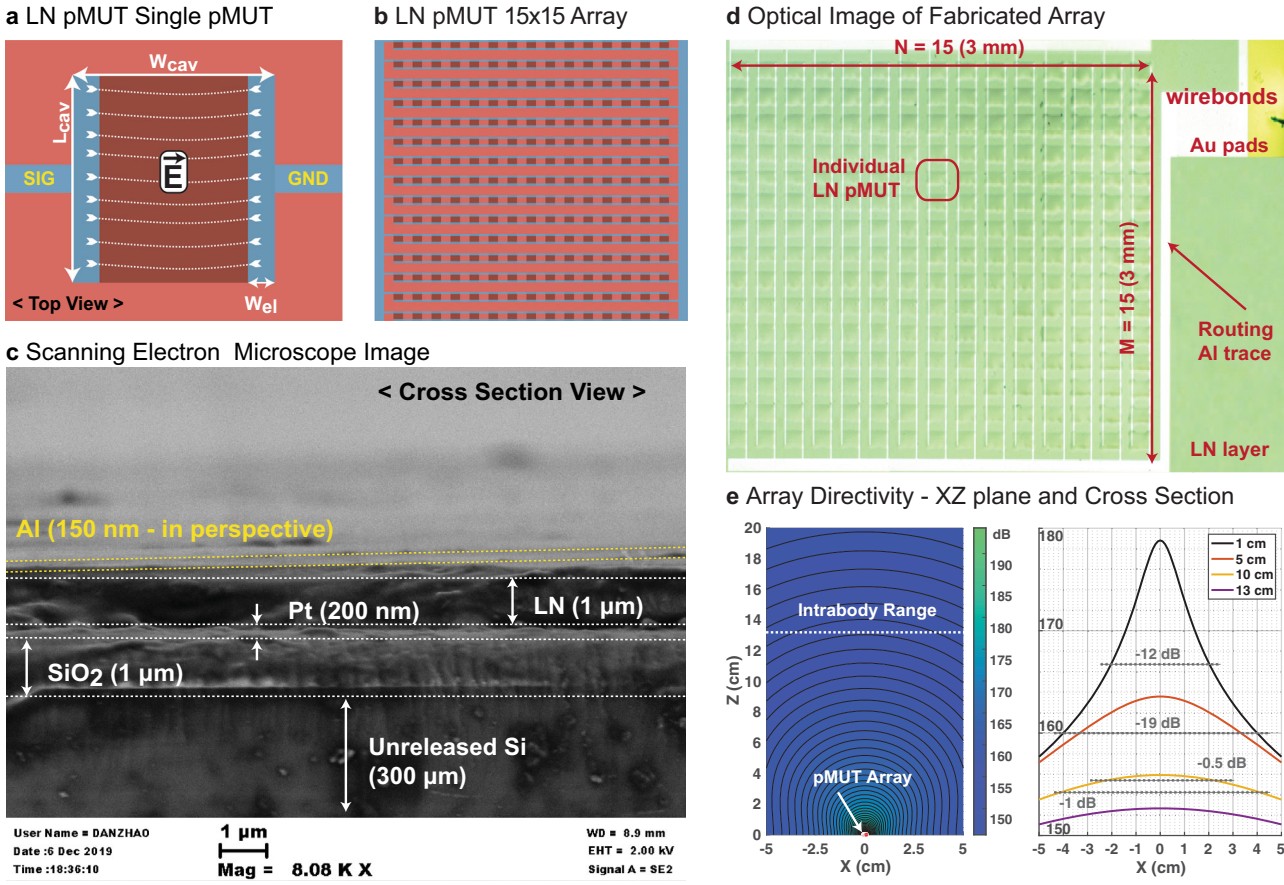

**Fig. 1 Lithium Niobate (LN) pMUTs. a** Design of a single element LN pMUT- dimensions of its cavity, electrodes, and bonding pads. **b** Design of a 15×15 LN pMUT array. **c** Cross section of LN pMUT with Scanning Electron Microscope (SEM). We can distinguish top electrode (Al), piezoelectric layer (LN), un-patterned bottom electrode (Pt), structural layer (SiO₂), and the unreleased substrate (Si). **d** Optical image of the fabricated 15×15 LN pMUT array. This is wire-bonded to a PCB and coated in Parylene for ultrasonic testing in a de-ionized water tank and then in a tissue phantom. The Parylene acts as a matching layer between the acoustic impedance of the pMUT membrane and the acoustic impedance of the medium, in this case, the tissue phantom. **e** Array directivity based on theoretical ultrasonic wave propagation in the XZ plane from the array (left figure) as well as the cross section in the X plane (right figure) at different distances from the pMUT array (1, 5, 10, and 13 cm).

included in Methods. This allows to recover the intensity of the received signal and to determine the bandwidth and maximum achievable distance. In Fig. 2b, we are showing the extracted bandwidth from the sound pressure level (SPL) received by the pMUT array. Finally, the communication setup through a body tissue phantom, implemented with the aid of two Universal Software Radio Peripheral (USRPs), is shown in Fig. 3. The communication results, representing the quality of the channel, are shown in Fig. 4 for different communication distances or depths. More details about the realized communication scheme in MATLAB Simulink and synthesized with the USRPs are shown in Methods.

**Lithium Niobate pMUTs**. The LN pMUTs are innovative since they harness, for the main resonance mode, a different piezo-electric coefficient compared to traditional pMUTs. The PZT, AlN, and ScAlN-based pMUTs consist of a piezoelectric thin-film sandwiched between a top and a bottom electrode in order to activate the $d_{31}$ piezoelectric coefficient[22]. The vertical electric field excites an in-plane displacement. Instead, in the case of the LN, we are taking advantage of a combination of piezoelectric coefficients (i.e., $d_{31}$ and $d_{11}$) that strongly couple the electric field to displacement[21]. These coefficients can be activated by electrodes that are on the same plane, thus the need of only one metal

layer. The full fabrication process is shown in Methods. In Fig. 1a we are showing the design elements of a single pMUT. This consists of the two activation electrodes, signal, and ground, of width $W_{el}$, and the releasing cavity that defines the vibrating membrane, of length $L_{cav}$ and width $W_{cav}$. More details concerning the array of pMUTs are shown in Methods. Additionally, in Fig. 1b we are showing the design of a $15 \times 15$ pMUT array. In Fig. 1c we are showing a Scanning Electron Microscope (SEM) image of a fabricated LN pMUT (unreleased). Here we can see the different layers such as the Si substrate, SiO₂ support layer, the Pt bottom electrode layer (not used for this device), the LN thin-film piezoelectric layer, and finally the top electrode Al (in perspective). Lastly, in Fig. 1d we are showing the optical image of the fabricated array. Here we can distinguish all the individual pMUTs and their released cavities (different shades of green), the routing traces connecting all the elements in parallel to combine the received and transmitted pressure, and the gold pads that are needed to wire-bond the array to a Printed Circuit Board (PCB). The whole array fits into a $3 \times 3$ mm² area, making it ideal as a miniaturized ultrasound antenna to implant into the human body alongside IMDs.

The displacement of the pMUT membrane is measured over the frequency range with a Digital Holographic Microscope (DHM) as shown in Fig. 2a (more details in Methods). The DHM creates a full 3D reconstruction of the membrane which allows for

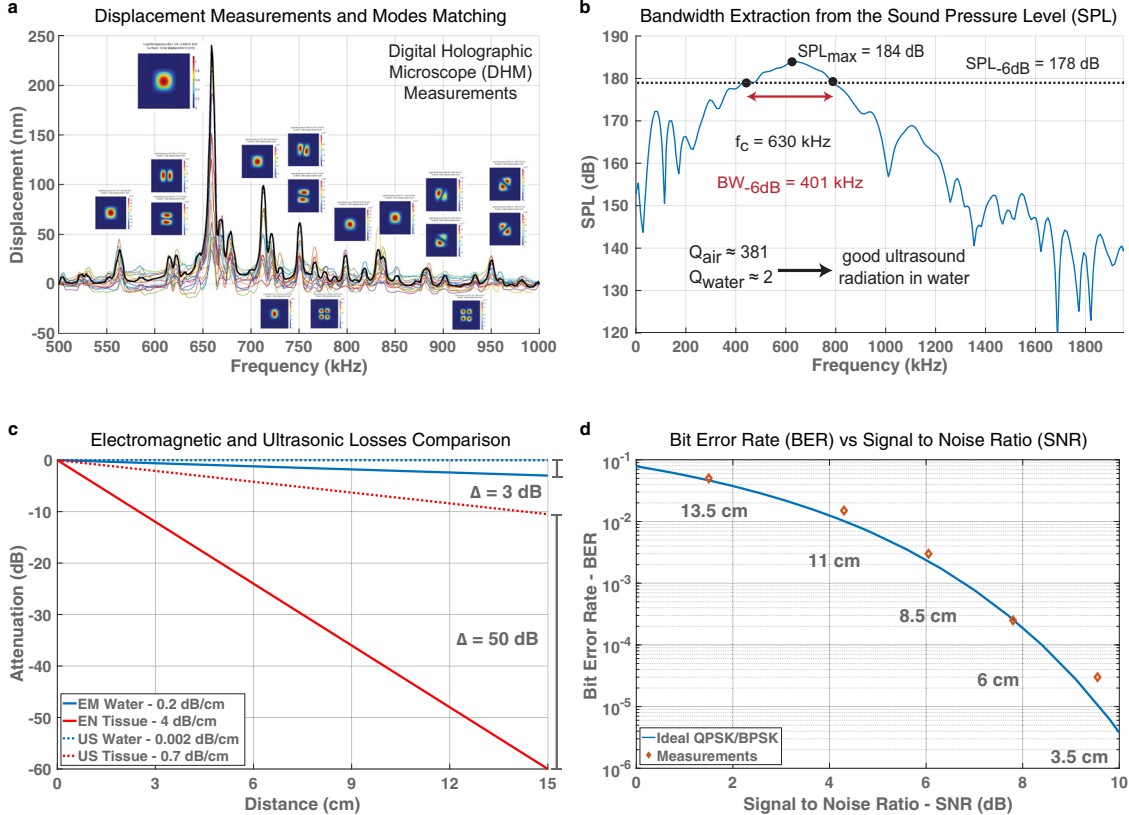

**Fig. 2 Testing. a** Peak displacement measurement of the LN pMUT device with a Digital Holographic Microscope (DHM) and modes matching in COMSOL Multiphysics. **b** Fourier transform of the received signal by the pMUT. This shows a bandwidth of $BW_{-6\,dB} \approx 401$ kHz and center frequency of $f_{water} \approx 630$ kHz (this is lower than the one in air due to the damping effect of the water). The low $Q_{water} \approx 2$ indicates that most of the energy is radiated, making the LN pMUT a good ultrasonic transmitter and receiver. This bandwidth is higher than pMUTs based on other piezoelectric materials such as AlN and ScAlN-36% as shown in Fig. 8c–e. **c** Attenuation of electromagnetic (EM) and ultrasonic (US) waves when considering both water and tissue phantom as propagation medium[18-20]. The implantation range considered is from 0 to 15 cm. In water, the EM waves have 3 dB more attenuation than US waves at 15 cm. However, this difference dramatically increases in tissue going up to 50 dB, making the US waves the ideal choice for intrabody communication. **d** Theoretical bit error rate (BER) vs Signal-to-Noise Ratio (SNR) plot for a Quadrature Phase Shift Keying (QPSK) communication link, which is the protocol that is going to be used in the real image transmission. The plot also presents measured BERs at several implantation distances that are going the be used in the intrabody setup. Some discrepancies are due to the misalignment of the transducers and implantation distance error, which can lead to lower SPL as shown in Fig. 1e.

precise measurement, both in terms of peak displacement and vibration mode. The figure shows the maximum membrane displacement of several pMUT samples and their mean value profile curve ("black-bold" curve). At this point, from the same curve, we can determine the main resonance frequency to be around $f_{res} = 660$ kHz and the maximum displacement $d_{max} = 240$ nm when applying an input voltage of $V_{in} = 3$ V, resulting in a displacement sensitivity of $S_{disp} = 80$ nm/V. Moreover, each major resonance peak is matched with a Finite Element Analysis (FEA) simulation in COMSOL Multiphysics (more details in Methods). The additional resonances are due to the strong anisotropic properties of the piezoelectric coefficient's matrix of the LN, which in this case play a key role in achieving a large operation bandwidth. This effect happens because the electric field generated by the top electrodes excites multiple modes through different piezoelectric coefficients[21], which are strong in multiple directions, as opposed to other materials that have strong coefficients only in one direction (AlN, ScAlN, and PZT). At this point, when submerging the pMUTs in a damping media such as tissue phantom or silicon oil, several adjacent resonance modes will merge together[23,24], resulting in an overall increased large bandwidth for the LN pMUTs. The details of the mode-merging modeling are described in Methods.

A time-domain measurement of the membrane displacement is performed with a Laser Doppler Vibrometer (LDV), as shown in Methods. The resonance frequency obtained from the LDV is $f_{res} \approx 699$ kHz and the peak displacement is $d_{max} \approx 88$ nm for an input signal of $V_{in} = 1$ V, resulting in a displacement sensitivity $S_{disp} \approx 88$ nm/V (confirming the 3D DHM measurement). While, on one hand, this technique allows to measure the displacement in only one point in the pMUT membrane, it also allows for a time-domain characterization. The time-domain approach allows to drive the pMUT membrane with several sine wave cycles ($N = 800$) and measure the ring-up and the ring-down of the membrane displacement, which is a function of the resonance quality factor[25], which results to be $Q \approx 381$. At this point, an array of $15 \times 15$ LN pMUTs is wire-bonded to a Printed Circuit Board (PCB) and covered in a thin Parylene layer. This layer has a dual function, one is protecting the wire bonds and the second is to act as a matching layer between the pMUT acoustic impedance and the operating medium impedance. At this point the LN array is submerged in a de-ionized (DI) water tank, to closely match the acoustic impedance of the human tissue and tested for its ultrasound transmission capabilities (more details in Methods). Finally, the Fast Fourier Transform (FFT) can be applied to the received pulse on this measurement, to extract the

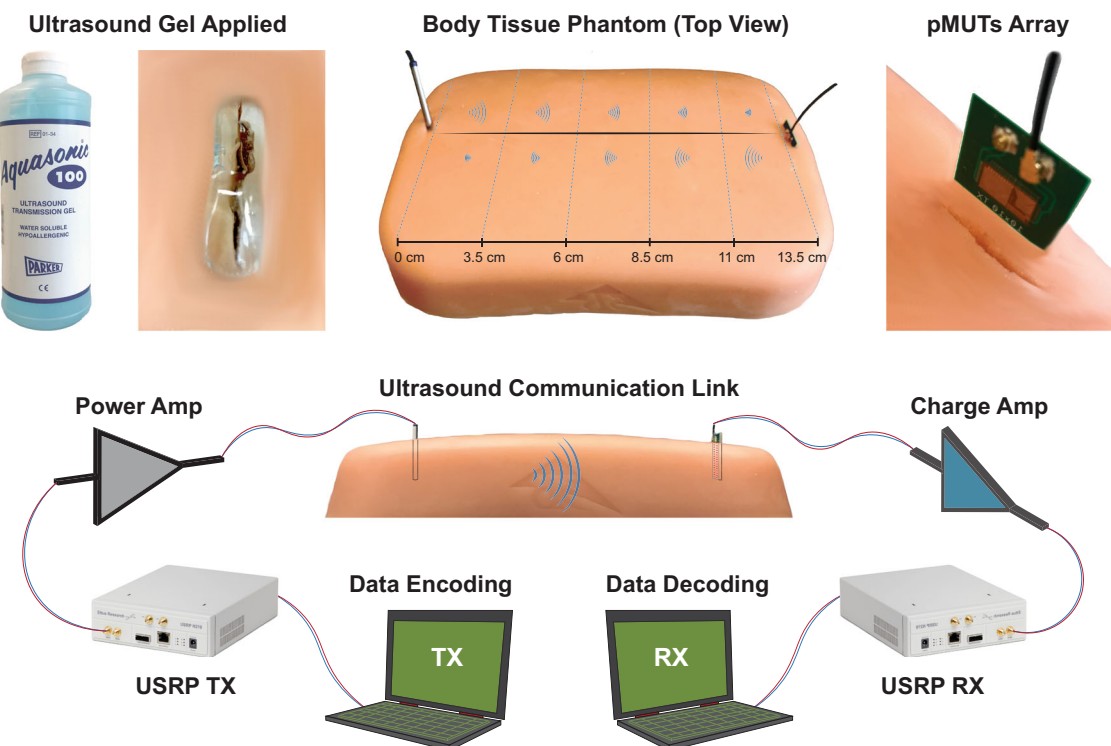

**Fig. 3 Intrabody setup.** LN pMUT arrays, wire-bonded to Printed Circuit Boards (PCBs) are implanted in a tissue phantom mimicking the human tissue properties. When implanting, the incised holes are filled with ultrasound gel to avoid air gaps that could reduce the intensity of the ultrasonic signal. Furthermore, the communication link is implemented with a transmitting and a receiving Universal Software Radio Peripheral (USRP). The signal of the TX USRP is amplified by an off-the-shelf power amplifier while the RX USRP is conditioned by an in-house custom-made charge amplifier designed to match the impedance of the LN pMUT array. In this setup, both TX and RX are LN pMUT arrays.

## Signal-to-Noise Ratio (SNR) vs. Bit-Error-Rate (BER) vs. Distance (D)

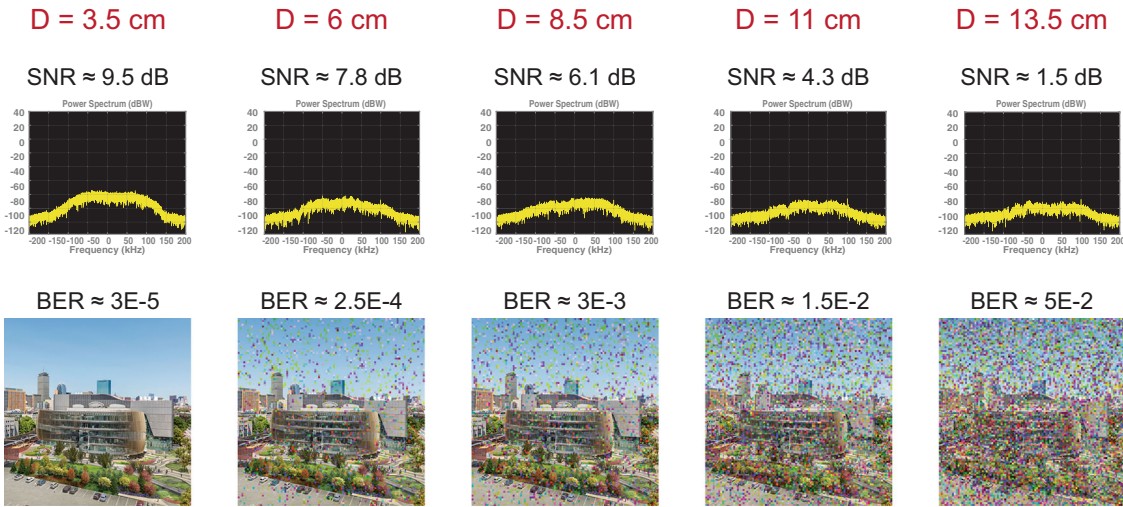

**Fig. 4 Communication results.** The ultrasonic channel is tested for Signal-to-Noise Ratio (SNR) and bit error rate (BER) at several implantation depths ranging from 3.5 cm up to 13.5 cm. Furthermore, the BER can be directly visualized with the transmitted data, the image of Northeastern University campus, in terms of lost or wrong pixel colors. Measurements are done between two LN pMUT arrays implanted in the tissue phantom shown in Fig. 3.

frequency domain response shown in Fig. 2b. From the graph, we can extract the operation bandwidth of the pMUT at −6 dB, which corresponds to a large bandwidth $BW_{-6\,dB} \approx 401$ kHz at a center frequency of $f_c \approx 630$ kHz. It is important to notice that the resonance frequency was lowered in the DI water due to the damping effect and at the same time, the modes shown in Fig. 2a are merged inside one large band. Similarly, we can extract the

quality factor in water which corresponds to $Q \approx 2$, which is two orders of magnitude lower than in air, indicating good ultrasound radiation in water or tissue-like media.

**Communication link**. Once the LN pMUT arrays have been characterized for ultrasound transmission, we are ready to set up an entire communication link that emulates as closely as possible

an intrabody communication scenario as shown in Fig. 3. The intrabody setup consists mainly of a body tissue phantom from 3B Scientific[26] that mimics the human body properties, such as propagation speed of the ultrasound waves, an average body density, and its corresponding acoustic impedance. An incision is performed on the phantom in order to implant the pMUT array receiver. At this point, in order to avoid air gaps around the implanted array, we apply an ultrasound gel[27], similar to the one used for ultrasound imaging, which gives continuity to the ultrasound transmission. Furthermore, the communication link is implemented with the use of Universal Software Radio Peripherals (USRP)[28]. This allows us to implement a communication scheme of choice in MATLAB Simulink (more details in Methods) and test it in real-time in the above described intrabody scenario. In a real-life scenario, the entire electronics can be implemented in CMOS circuitry or miniaturized Field Programmable Gate Arrays (FPGA). While other groups have demonstrated miniaturized electronics for intrabody communication[29,30], this goes beyond the purpose of this paper. The main focus of this work is to demonstrate the high performance of the LN pMUT arrays, in an intrabody scenario, in terms of large bandwidth and long intrabody communication range. In order to take full advantage of the large 400 kHz bandwidth provided by the LN pMUTs, we choose a Quadrature Phase Shift Keying (QPSK) modulation scheme as the communication protocol. Here we are choosing to transmit the row pixels of an image over the ultrasound link and then the information is serialized into a bitstream. More details about the encoding and the QPSK modulation are included in the Methods. At this point, the bitstream can be fed directly into a QPSK modulator object provided in MATLAB Simulink which will interface with the USRP Software Defined Radio (SDR) transmitter. The SDR transmitter is in charge of up converting the modulated data from baseband (DC center frequency) up to the RF frequency corresponding to the central frequency of the pMUT

array transmitter $f_c \approx 630$ kHz. Finally, once the data payload is ready, we can proceed with a decoding MATLAB script (the reverse procedure used to encode) and re-assemble the received data into an image. At this point, we are able to compare the received image with the originally transmitted one in terms of BER at several communication distances or implanted device penetration depths, which will degrade with lower SNR at longer distances. By choosing an image as transmitted data we can visually interpret on-the-fly the quality of the ultrasonic channel in terms of lost pixels ("black") or degraded pixels ("un-real colors").

Once the communication setup is ready to transmit and receive, testing can be performed. The main test consists of characterizing the quality of the ultrasonic channel in the tissue phantom at different distances, starting from a minimum of $D_{min} = 3.5$ cm and a maximum distance in the phantom of $D_{max} = 13.5$ cm. The results are shown in Fig. 4. The Signal-to-Noise Ratio (SNR) obtained for such communication depth range has a maximum of $SNR_{max} = 9.5$ dB and reaches a minimum of $SNR_{min} = 1.5$ dB as shown in the power spectrum of the QPSK modulation. Under these SNR conditions, we are computing the amount of pixel errors when receiving the encoded image, a metric which is commonly known as BER. In order to have a direct interpretation of the BER and thus of the quality of the ultrasonic channel based on the LN pMUT arrays, in Fig. 4 we are showing the decoded image for each distance. Here we can appreciate how some pixels got "lost" or have the wrong information resulting in abrupt color changes compared to other adjacent pixels. On one hand, the BER had a minimum of $BER_{min} = 3 \times 10^{-5}$ where real-time data transmission can be directly implemented on the communication channel without additional coding or electronics[31]. On the other hand, the BER has a maximum of $BER_{max} = 5 \times 10^{-2}$ which is the limit for being able to correctly reconstruct the original information by implementing bit error detection and correction algorithms[32].

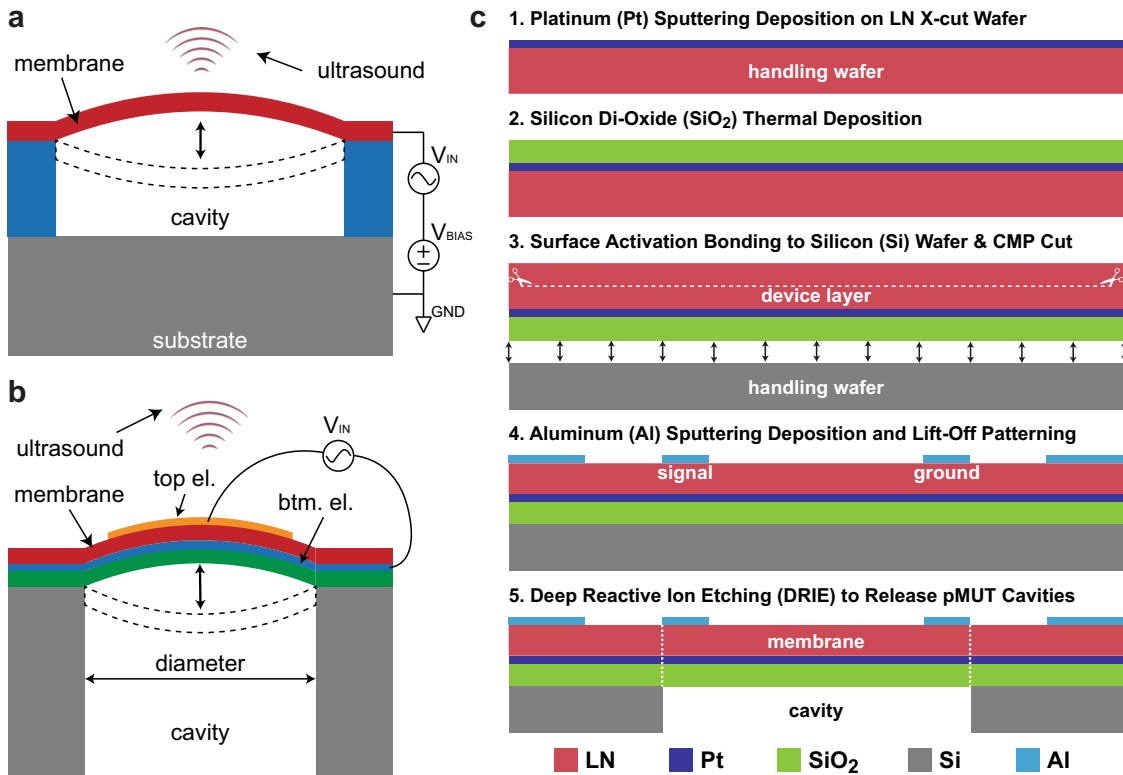

**Fig. 5 MUT categories and fabrication of LN pMUTs. a** Working principle of a cMUT and the main composition layers. **b** Working principle of a pMUT and the main composition layers. **c** Step-by-step fabrication process of the LN pMUTs.

The $BER_{max}$ sets the limit to the implantation depth of the IMD in order to maintain reliable ultrasonic data transmission. Additionally, the experimental BER vs SNR is plotted in the initial theoretical curve for a QPSK link in Fig. 2d, showing the logarithmic decrease of the communication errors while the signal level decreases at longer implantation depths. These results are promising toward allowing a broad implantation range $D_{range} = 3.5–13.5$ cm which enables the implantation of all three categories of IMD discussed in the introduction and at the same time offers a large communication bandwidth of BW = 400 kHz. Ultimately, this visual interpretation of the ultrasonic channel quality can be useful for applications such as scanning for multiple IMDs in order to find an optimal location for the external transmitter[33].

In conclusion, given the results in terms of large bandwidth and deep implantation range, we believe that the LN-based pMUT technology has the potential to significantly improve the wireless communication links for implanted medical devices for real-time monitoring. The LN piezoelectric thin film shows promising insights on how to achieve a large band thanks to the combination of spurious modes under the damping of the acoustic medium such as water and tissue. This leaves open the research to explore different actuation designs of the pMUT with the X-cut LN layer, different device orientations, as well as exploring different crystal cuts, such as Y- and Z- cuts.

## Methods

**Micromachined Ultrasonic Transducers (MUT) categories and fabrication of LN pMUTs.** The first category of transducers is the capacitive MUTs (cMUTs), which generally consist of a suspended membrane on top of a silicon substrate that is actuated (i.e., put in vibration mode) by an electric field as shown in Fig. 5a. This electric field is generated by applying a high DC bias to the membrane (it helps make the membrane more compliant or softer) and an AC driving voltage signal that sets the membrane into motion at its resonance frequency. When in motion, the membrane will displace the particles in the surrounding medium and generate ultrasonic waves that can propagate in the far field of the device and be used as carrier for an ultrasonic communication link.

The second category is the piezoelectric MUTs (pMUTs), which consist of a membrane that moves at its resonance frequency, generating ultrasonic waves as shown in Fig. 5b. The simplest pMUT structure consists of a thin-film piezoelectric membrane, for example, made of Aluminum Nitride (AlN), that is sandwiched between two electrodes (i.e., top and bottom electrodes). When applying a driving AC voltage signal to the electrodes, the AlN will start expanding and contracting in the lateral dimension due to the inverse piezoelectric effect. Since the membrane is clamped and suspended on top of a cavity, this will act as a boundary condition that will force the membrane to vibrate in the Z-axis during its expansion and contraction in the lateral dimension.

In this work, we are interested in LN-based pMUTs, and their step-by-step fabrication process shown in Fig. 5c. The first part of the process starts with a thin film X-cut LN wafer of 300 μm on which a Pt layer of 200 nm is sputtered (step 1). Following, a $SiO_2$ layer of 1 μm is added through chemical vapor deposition (step 2). The presence of this layer shifts the neutral bending axis of the pMUT device to the middle of the piezoelectric layer, which maximizes the membrane displacement. So far, the LN acts as the handling wafer for the Pt and the $SiO_2$ layers, but the LN needs to act as a device layer instead. Thus, it needs to be transferred to another handling wafer. In order to achieve this passage, the LN wafer is flip bonded to a Double Side Polished (DSP) Si wafer of 300 μm thickness,

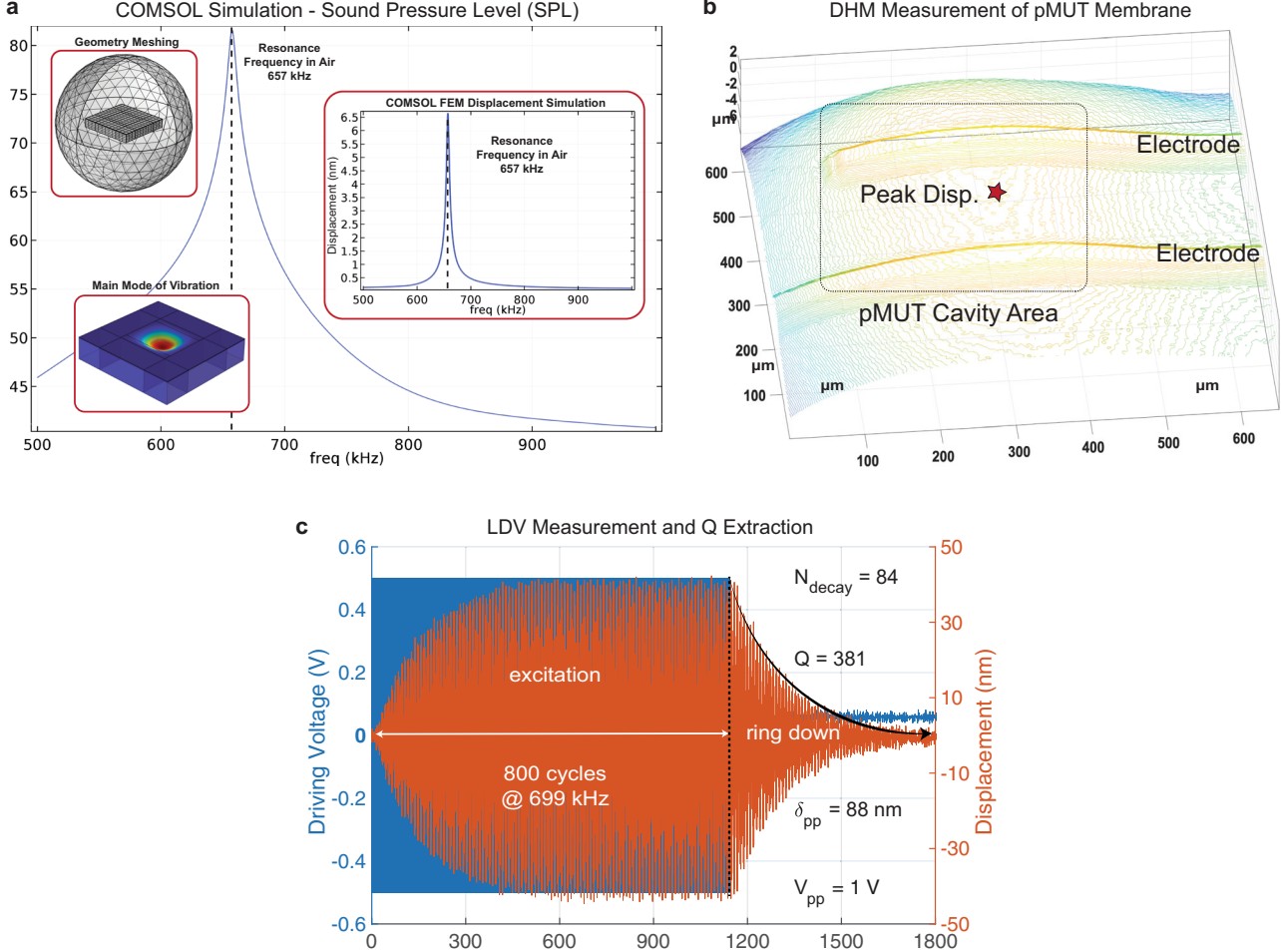

**Fig. 6 LN pMUT simulation and air measurements. a** COMSOL Multiphysics Finite Element Analysis (FEA) simulation of an LN pMUT. **b** DHM measurement of the pMUT membrane displacement and its 3D reconstruction for mode analysis. **c** Quality factor extraction from the displacement of the pMUT membrane with the ring-down method.

through a surface-activated bonding technique (step 3). In the same step, the LN wafer is reduced to a device thickness of 1 μm through a Chemical and Mechanical Polishing (CMP) process. Now the LN is a device layer, and the Si wafer acts as a handling wafer. In the second part of the fabrication process, patterning masks are used to lithographically implement the design of the desired pMUTs, including both single-elements and array layouts. In particular, there is an electrode definition mask and a cavity releasing mask. The first one will define the 150 nm Al top electrode that is electron-beam sputtered and shaped through a lift-off process (step 4). Finally, the second mask defines the pMUT cavities during the releasing process (step 5). This step consists of a Deep Reactive Ion Etching (DRIE) process to etch straight trenches (i.e., cavities) from the back of the Si wafer and stop on the SiO2 layer, thus releasing the pMUT membrane.

**COMSOL simulation and air measurements of fabricated devices**. To simulate the pMUTs, COMSOL Multiphysics is used to generate a 3D model of a single LN pMUT and to run a Finite Element Analysis (FEA) to find its behavior in the frequency domain. Besides providing several physics domains for the models, the advantage of using COMSOL is that is also provides coupling modules between these different domains. In fact, to simulate a pMUT element we make use of two multiphysics modules: the piezoelectric module, which couples the electrical domain with the mechanical one, and the ultrasonic module, which couples the mechanical domain with the sound-pressure domain in different materials. This simulation tool allows us to set up the application medium such as air and tissue-like media (oil, water, tissue phantom, etc.) and select a particular thin-film layer as the piezoelectric layer. This allows to select a LN cut (i.e., crystal orientation and piezoelectric coefficient matrix), in our case the X-cut, and the angle of orientation at which the electric field is applied. In Fig. 6a we are showing the sound pressure level (SPL) generated by the pMUT while sweeping the operation frequency. Here we are detecting a resonance frequency of 657 kHz in the air where the SPL is maximum. Finally, the three inserts show the geometry mashing, the main mode of vibration, and the membrane displacement over frequency.

Once the devices are fabricated, these are characterized in the air with a Digital Holographic Microscope (DHM) to measure the full 3D displacement of the pMUT's membrane as shown in the reconstruction in Fig. 6b and detect the mode shape. Here we measured a main resonance of $f_{air} \approx 669$ kHz (close match to simulation) and an average peak displacement sensitivity of $S_{disp} \approx 93$ nm/V. Following, the peak displacement is measured with a Laser Doppler Vibrometer (LDV) while driving the pMUT with $N = 800$ sine wave cycles at the resonance frequency in air. This allows to extract its quality factor $Q$ with the ring-down

technique. This consists of counting how many cycles it takes to halve the displacement amplitude once the driving signal is turned off, and then multiplying this number by 4.53 to find the quality factor $Q$. In our case, the ring-down decay takes $N_{decay} = 84$, as shown in Fig. 6c, thus $Q_{air} \approx 381$.

**Modes merging in air vs water/tissue**. The LN pMUTs present interesting piezoelectric properties for which multiple resonance modes can be activated around the main resonance frequency. With a frequency sweep on the DHM, we can detect the additional resonance modes based on the shape of the 3D displacement of the membrane as shown in Fig. 6b. The modes are then extracted from the measurement and fitted to a Butterworth–Van Dyke (BVD) model based on their frequency, quality factor, coupling factor, and peak displacement, as shown in Fig. 7a. Since these measurements are performed in air the quality factor of each resonance is high, making them very distinctive from one another. This is due to the high acoustic impedance mismatch to the air which prevents the ultrasonic radiation and keeps most of the energy on displacing the pMUT's membrane. Similarly, the air has also a low density, thus the damping of the displacement is low as well. Following, Fig. 7b shows the simulated combined SPL at the output of the LN pMUT surface based on the above measurements. As it can be noticed, the peaks do not merge together but rather form multiple fractional bands, because of the high-quality factor of each peak in air.

Finally, we model what happens to the peak displacements of all the resonance modes and to the combined output SPL of the LN pMUT when exposed to a denser external load, such as water or a tissue phantom. First, in Fig. 7c the displacement attenuation of each resonance peak and the lowering of their corresponding quality factor can be noticed. This is due to the more efficient radiation of the energy from the pMUT surface into the medium by generating ultrasonic waves. Moreover, given the high density of the new medium, this also has a damping effect on the displacement amplitude of each resonance peak, allowing them to merge into a single output pressure band as shown in Fig. 7d. In this case, the resulting large bandwidth of $BW_{-6 dB} \approx 400$ kHz is a very close match to the measured results in Fig. 2b with a reference hydrophone at a radiation distance of 3.3 cm.

**Ultrasonic measurement setup, bandwidth extraction, and comparison**. A 15×15 LN pMUT array is coated with Polydimethylsiloxane (PDMS) and submerged in a de-ionized water tank and tested for ultrasonic transmission as shown in Fig. 8a. This shows a record bandwidth of $BW_{-6 dB} \approx 401$ kHz and a center frequency $f_{water} \approx 630$ kHz. In addition, $Q_{water} \approx 2$ is lower than in $Q_{air}$, implying that most of the input energy is radiated into the medium, making the LN pMUT ideal for underwater

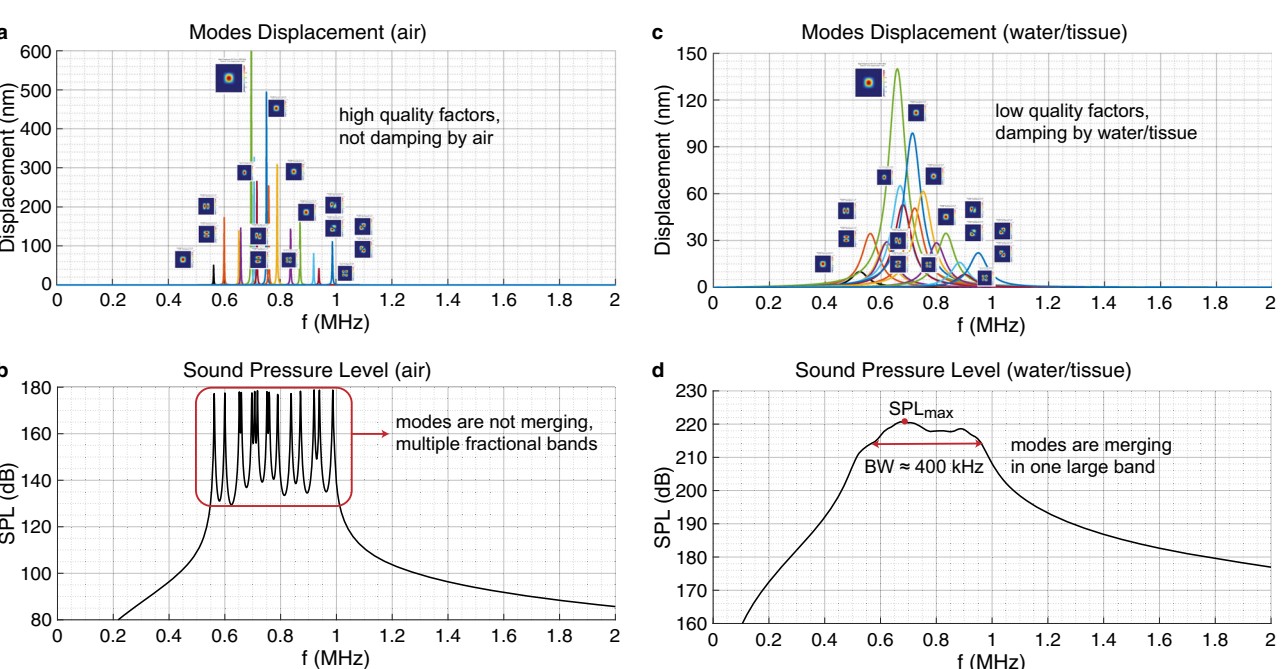

**Fig. 7 Modes merging in air vs water/tissue. a** LN pMUT 3D membrane displacement measurement and peak detection in air. Modes are matched to COMSOL Multiphysics simulation based on their shape. Here we notice that the peaks are separated from each other due to a high-quality factor in air. The different colors represent different modes extracted from the measurement in air from Fig. 2a. **b** Simulated combined sound pressure level (SPL) of the device in air. Here we can notice that the different peaks are not merging, instead we have the formation of multiple fractional bands. **c** Displacement modeling of the resonance peaks under the damping effect of the water and tissue medium. Here the quality factor is getting lower and closer to one another. The different colors represent different modes extracted from the measurement in air from Fig. 2a and then modeled under the damping effect of the water, by decreasing the quality factor. **d** Modeling of the computed output SPL resulting in a unified large band of BW ≈ 400 kHz, matching very closely with the reference hydrophone measurement.

and intrabody communication. In Fig. 8b we are showing the driving signal ("blue" curve) of a hydrophone and the received pulse by an LN pMUT array ("orange" curve). The hydrophone is excited with a high-intensity pulse to emulate a Dirac pulse, $V_{Dirac} = 9$ V and $f_{Dirac} = 5$ MHz, which allows to extract the step response of the system. It is interesting to notice that the received signal is delayed by 21 μs at a distance of 3.3 cm, which allows us to estimate the sound velocity in the water to be $c \approx 1450$ m/s. Finally, the received pulse has an intensity of $V_{RX} \approx 4.2$ mV.

At this point, we can compare pMUTs based on other materials, such as AlN as shown in Fig. 8c and ScAlN doped at 36% as shown in Fig. 8d, and then compare it to the bandwidth obtain in this work for LN pMUTs as shown in Fig. 8e. The impulse response is measured in a water tank as in Fig. 8b and FFT is applied for bandwidth extraction at −6 dB from the peak as shown in Fig. 2b. The LN-based pMUTs of this work show the highest bandwidth of 401 kHz, while the 36% Sc-doped AlN shows large bandwidth of 375 kHz as well, which makes it a good material to explore. However, a disadvantage of the Sc is the reproductivity of the sputtering process, while the advantage of the LN is that during the bonding process to the substrate it maintains the same crystal structure and piezoelectric properties, hence making it more reliable and predictable.

**Data serialization into a bitstream and modulation scheme implementation.** First, a raw image of $100 \times 100$ pixels is serialized in MATLAB to create a bitstream for the communication scheme as shown in Fig. 9a. Each pixel consists of an RGB

vector of three integers (0–255) that can be converted into an 8-bit string, for a total of 24 bits for the vector. At this point, all the pixels are concatenated in a bitstream resulting in a total raw data of $Data_{RAW} = 240$ kbits. Then, the bitstream is encoded with a Quadrature Phase Shift Keying (QPSK) modulation, which allows to encode 2 bits per second as shown in Fig. 9b. The modulation is done asynchronously eliminating the need for a clock. On the other hand, there is the need to add an overhead to the raw data in order for the receiver to detect it. This increase in data length is of ~10%, resulting a final $Data_{QPSK} \approx 264$ kbits. Finally, the QPSK data is up-converted by the USRP at the operation frequency of the pMUT array and transmitted through a tissue phantom that mimics the human tissue properties as shown in Fig. 9c. The pMUT array bandwidth BW ≈ 400 kHz, translating in a data rate ≈ 800 kbits/s, meaning that the time to transmit one image or frame is $T_{TX} \approx 0.33$ s. On the receiving side of the intrabody ultrasonic transmission link, the signal is down-converted to baseband by another USRP and sampled at twice the bandwidth for perfect reconstruction (Nyquist theorem). The signal requires constant frequency and frame synchronization as discussed in the paper. At this point, the data bitstream is demodulated from the QPSK scheme and re-assembled in an RGB pixels matrix.

The generated bitstream can be fed directly into a QPSK modulator object provided in MATLAB Simulink which will interface with the USRP Software Defined Radio (SDR) transmitter. The SDR transmitter is in charge of up converting the modulated data from baseband (DC center frequency) up to the RF frequency corresponding to the central frequency of the pMUT array transmitter $f_c = 630$ kHz, as shown in Fig. 9d. The encoded bitstream is received by a pMUT

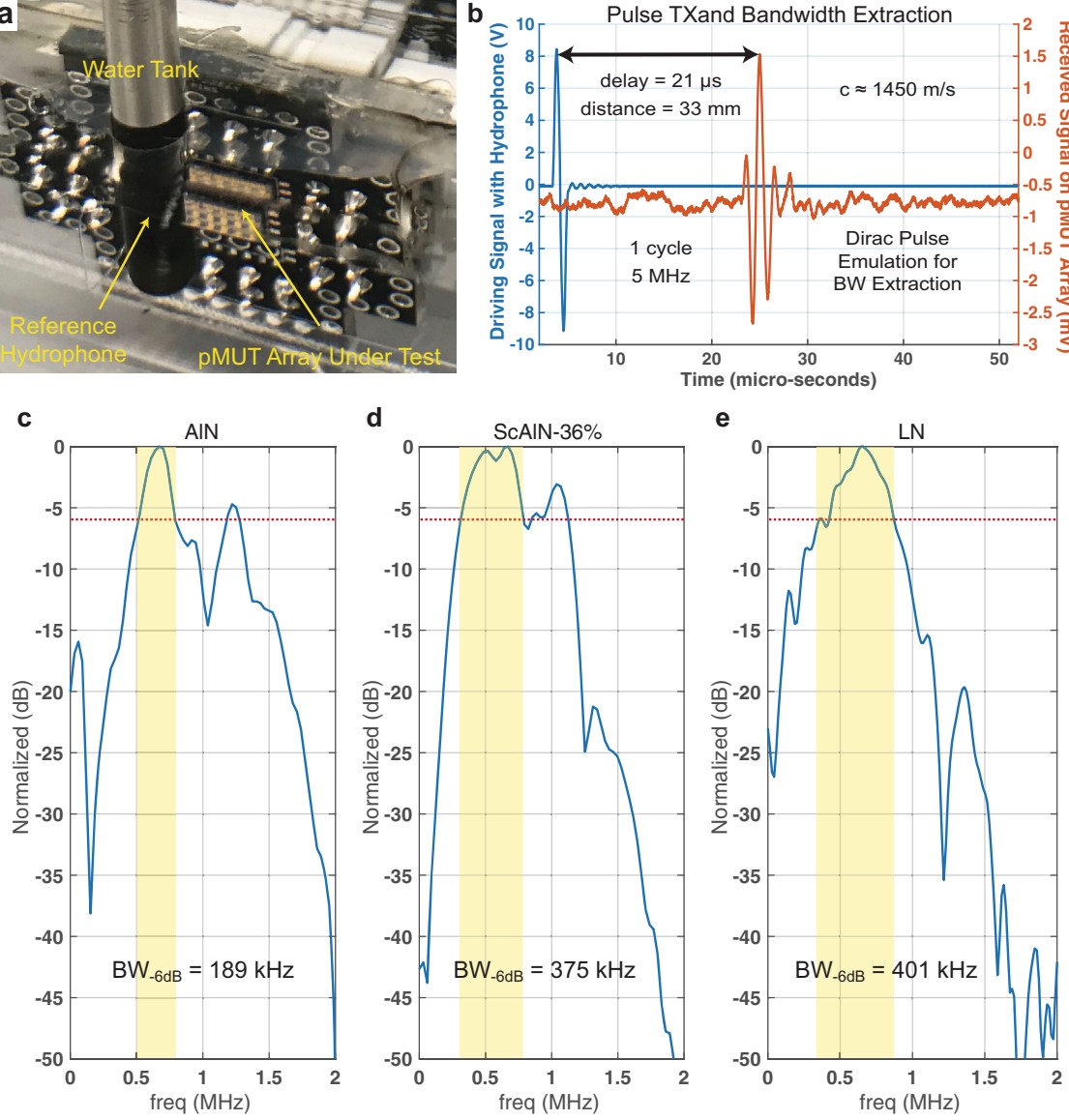

**Fig. 8 Ultrasonic setup, bandwidth extraction, and comparison. a** Measurement setup for ultrasonic transmission between a commercial hydrophone (Teledyne) and the LN pMUT array submerged in a De-Ionized (DI) water tank. **b** Dirac pulse transmission to extract the pMUT bandwidth (BW) with a Fast Fourier Transform (FFT). **c** BW of an AlN pMUT. **d** BW of a ScAlN-36% pMUT. **e** Bandwidth of the LN pMUT in this work, at the same frequency of AlN and ScAlN-based pMUTs for comparison.

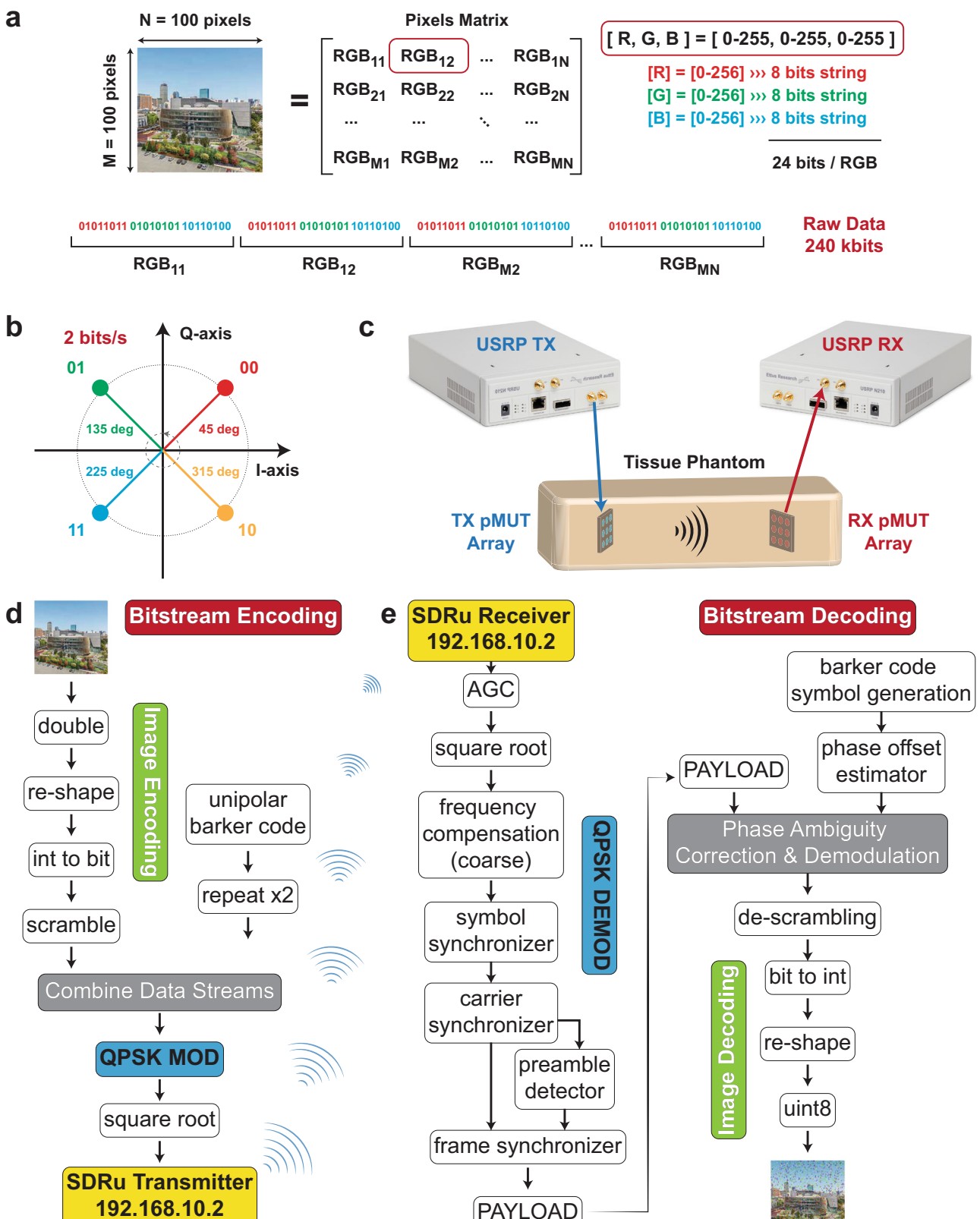

**Fig. 9 Data encoding and communication scheme. a** Image encoding. **b** QPSK constellation. **c** Testing setup. A power amplifier was used to boost the output voltage of the TX USRP and a charge amplifier was used to detect the incoming signal at the RX USRP. Both TX and RX are pMUT arrays. **d** A 100 × 100 pixels image of the Northeastern University campus is serialized into a bitstream in MATLAB. Further, this is encoded with a QPSK modulation and transmitted over an USRP in Simulink. **e** The transmitted data is received over the ultrasonic channel and synchronized with a second USRP in Simulink. Finally, the data is decoded from the QPSK modulation and re-shaped into a 2D RGB image.

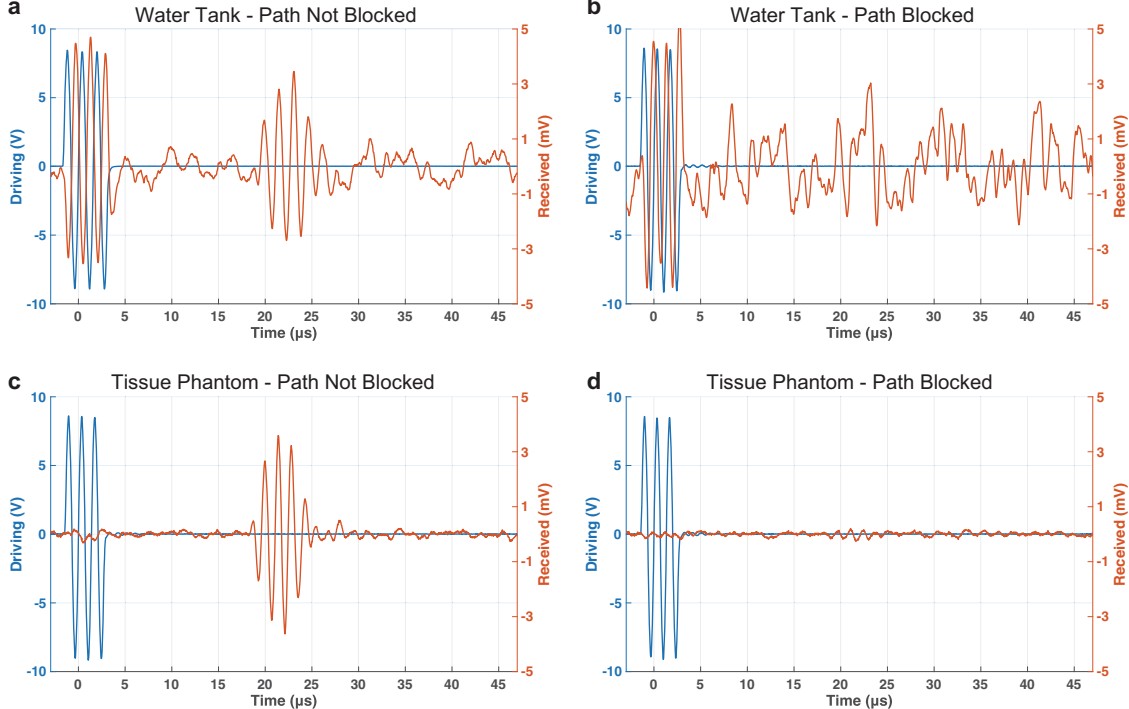

**Fig. 10 Time-domain comparison of pulse transmission. a** Pulse transmission in water with path not blocked. Both ultrasonic pulse and capacitive coupling signals are present. **b** Pulse transmission in water with path blocked, only capacitive coupling is present. **c** Pulse transmission in tissue phantom, only the ultrasonic pulse is present, and the capacitive signal disappeared. **d** Pulse transmission in tissue phantom with path blocked, the ultrasonic pulse disappears as well.

array receiver with the same central frequency as shown in Fig. 9e. At this point, since the receiver and the transmitter don't share a common clock, the major issue is synchronizing the SDRs at the two ends of the communication link. For this reason, we are using three standard synchronization blocks for the QPSK modulation, such as frequency offset compensation, symbol, and carrier synchronization. This will allow to maintain a stable ultrasound link and even stream the data in real-time.

**Time-domain comparison to avoid capacitive coupling**. When in water, we have that the driving signal couples directly into the received signal through a capacitive coupling effect, while the received ultrasonic pulse is received with a delay of 20 μs, which indicated a distance of around 3 cm as shown in Fig. 10a. Then the ultrasonic path is blocked, and the capacitive coupling is still present in water as shown in Fig. 10b. At this point, when repeating the same experiments in a tissue phantom, the capacitive coupling is not present anymore as shown in Fig. 10c. In this plot, we can see the ultrasonic pulse delayed by 20 μs at around 3 cm without any of the driving signal leaking into the received signal. Additionally, in Fig. 10d we are blocking once again the direct path in between the TX and RX and the ultrasonic pulse is not present anymore. This shows us that by using a tissue phantom in the experimental setup we can avoid capacitive coupling in the ultrasonic pulse and hence implement a reliable communication link.

## Data availability
Data are available upon request.

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

## Acknowledgements
This work was supported by the National Science Foundation (NSF) through the following grants: 1618731, 1726512, and 1763964.

## Author contributions
F.P and M.R. conceived the idea; F.P. designed the device and the experimental setups and performed modeling and simulations; F.P and B.H. fabricated and tested the devices and performed the experiments and analyzed the data; M.R. initiated, coordinated, and supervised the research. F.P., B.H., and M. R. contributed to the preparation of the manuscript.

## Competing interests
The authors declare no competing interests.
