## [Peer Review File · Nature Communications]

REVIEWER COMMENTS

Reviewer #1 (Remarks to the Author):

The paper demonstrates that use of thin film LN for PMUTs. In this case CMP was used to achieve 1 μ m LN, virtually eliminating most of the thick LN. The process is novel, and will provide future designers to utilize LN for PMUT devices. An ultrasonic data link was demonstrated in a tissue phantom over a distance of 13.5cm. The bandwidth of the transducer being 400kHz indicates a relatively high communication rate.

Comments:

1. "RF antennas operating at much higher frequency but having a dramatic increase of attenuation in tissue." - This needs to be justified." In order to justify this, a curve of RF absorption vs. US absorption would be useful.
2. A theoretical model of absorption vs measured BER may solidify the understanding of the communication link. Is the coupling to liquid an issue - i.e. the impedance match between the transducer and the tissue?
3. It would be easier to conduct the commlink experiments in a water tank at greater distances to confirm the link budget, as the loss in water at the transducer frequencies are well documented.
4. It will be useful to compare to AIScN and AIN in the context of bandwidth availability, and also to CMUTS, where the claim is often made of high bandwidth. Thinning down a thick LN wafer to 1 μ m maybe cost challenging.

Reviewer #2 (Remarks to the Author):

The authors present an ultrasound transducer made from lithium niobate that has resonant frequencies of ~400Khz. The transducer is used for ultrasonic intrabody communication.

The paper demonstrates a good application of lithium niobate material in a piezoelectric transducer configuration. It would be more suitable for an engineering journal e.g. IEEE as the focus is more on the systems development, implementation and testing of the transducer. The use of lithium niobate for ultrasonic transducers itself is not new, with several commercial products already available and patents dating more than a decade old.

Reviewer #3 (Remarks to the Author):

The paper describes the design and construction of piezoelectric micromachined ultrasonic transducer arrays and experiments to explore the feasibility of high data rate ultrasonic communication based on these devices. The subject is of definite interest to the field and presents a promising approach for implanted devices requiring high data throughput. However I would like to see clearer explanation of how this work advances that published in Herrera, B., Emrehan Demirors, G. Chen, R. Guida, Flavius V. Pop, Neil Dave, C. Cassella, T. Melodia and M. Rinaldi. "PMUT-BASED HIGH DATA RATE ULTRASONIC WIRELESS COMMUNICATION LINK FOR INTRA-BODY NETWORKS." (2018) and other related publications from the same team.

The work on the design and construction of PMUT devices is well described, reproduceable and a valuable contribution. In water testing of frequency response and impulse response (using hydrophones) is convincing but I find it strange that no results are presented on the directivity of the device. I would expect to see measurements of the radiated beam pattern in at least one plane. This is clearly an important issue for practical application in implanted devices - how well aligned do the Tx and Rx arrays need to be to

achieve the quoted performance?

The communication performance is an area with some significant weaknesses. It would be more convincing if the channel impulse response and/or time domain signals for each range on the tissue phantom were presented (similar to their water tests), as well as the spectra. Not only does this help the reader to see if the communication channel is noise or multipath limited but appropriate propagation delay also confirms that the principle signal path is acoustic rather than electromagnetic crosstalk. (It is challenging to 100% eliminate EM coupling when using MHz frequency ultrasonic signals in an experiment with long cables, open PCB tracks and an unshielded tissue phantom). The spectrum of the signals remains remarkably flat over the range 3 - 13 cm when I would expect to see the differential attenuation between the low and high frequency band edges to be more visible, especially at the longest range. Again, seeing time domain signals would confirm that signal coupling was always through the expected mechanism.

Also the quoted bit error rate of 10^{-1} at the maximum range is very high and, although appropriately designed error correction codes code make this useable, this would be at the expense of a large reduction in data throughput since a quite low code rate would be required. Raw pixel data (allowing pixel errors) would be unlikely to be used in a practical imaging system and compressed images with error control coding are much more likely. These results need to be described in the context of delivering high integrity data (low error rate $< 10^{-5}$).

Reviewer #1

The paper demonstrates that use of thin film LN for PMUTs. In this case CMP was used to achieve 1 μ m LN, virtually eliminating most of the thick LN. The process is novel and will provide future designers to utilize LN for PMUT devices. An ultrasonic data link was demonstrated in a tissue phantom over 13.5 cm. The bandwidth of the transducer being 400 kHz indicates a relatively high communication rate.

Comments:

1. "RF antennas operating at much higher frequency but having a dramatic increase of attenuation in tissue." - This needs to be justified." In order to justify this, a curve of RF absorption vs. US absorption would be useful.

We thank the reviewer for pointing this out. We agree that showing the different attenuation rates of the electromagnetic (EM) waves compared to the ultrasonic (US) waves will strengthen and justify our claim. For this reason, we added the plot in Figure 2c showing the attenuation of the EM and US waves from 0 to 15 cm implantation depth, when considering both water and tissue phantom as propagation medium. In water, the EM waves attenuate with a rate of 0.2 dB/cm while the US waves attenuate at a rate of 0.002 dB/cm; even though the US waves attenuate much less, it only introduces a different of 3 dB at 15 cm, which is beyond the maximum range of our experiment. However, when considering the tissue phantom, the EM waves attenuate at a rate of 4 dB/cm while the US waves attenuate only at a rate of 0.7 dB/cm; this introduces a different of 50 dB at 15 cm when operating in tissue phantom, which is closer to the human body in terms of attenuation and acoustic impedance, hence making the US waves a better fit for intrabody communication¹⁸⁻²⁰ such as in our application with the LN pMUTs.

2. A theoretical model of absorption vs measured BER may solidify the understanding of the communication link. Is the coupling to liquid an issue - i.e., the impedance match between the transducer and the tissue?

We thank the reviewer for this comment. In addition to the absorption plot we added for the previous comment, we also added Figure 2d showing the theoretical Signal-to-Noise Ratio (SNR) vs Bit-Error-Rate (BER) plot for a Quadrature Phase-Shift Keying (QPSK) communication link, together with the measurement results at different distances (3.5, 6, 8.5, 11, and 13.5 cm). This will help showing the reader how the BERs from Figure 4 are following the same trend as the theoretical curve. We speculate that some discrepancies are due to misalignment of the transmitter and receiver as well as some error in the implantation distance due to the incision and filling with ultrasonic gel (top left corner in Figure 3). In particular, the misalignment can lead to lower Sound Pressure Level (SPL) due to the directivity of the array (a new figure has been added as Figure 1e).

Regarding the second part of the question, to improve the matching between to the acoustic impedance to the medium, two methods have been implemented. The first is to coat the array with a thin Parylene layer (< 1 mm) which acts as an intermediate impedance medium, reducing the acoustic reflections at the interface between the tissue phantom and pMUT array. Additionally, this method also protects the gold wire bonds from the array to the Printed Circuit Board (PCB) as shown in Figure 1d. The second method is to fill all the air gaps from the implantation incision with ultrasonic gel, which avoids reducing the SPL due to traveling in air pockets (top left corner of Figure 3).

Array Directivity - XZ plane and Cross Section

3. It would be easier to conduct the commlink experiments in a water tank at greater distances to confirm the link budget, as the loss in water at the transducer frequencies are well documented.

We thank the reviewer for bringing up the measurements in the water tank. While some of these experiments have been conducted in the water tank, such as bandwidth extraction in Figure 2b and pulse detection in Figure 13, none have been conducted with the communication link implemented. The main reason is because in water we encounter capacitive coupling between the transmitter and receiver, which adds a clear interference pulse to the receiver equal to the driving signal. For bandwidth extraction purposes, if the ultrasonic pulse is sufficiently delayed (enough distance between TX and RX), then it will not overlap with the capacitive signal, the driving signal can be decoupled and windowed out and perform the Fast Fourier Transform (FFT) only on the ultrasonic pulse, which is delayed compared to the capacitive signal. We tried to eliminate this capacitive effect in water by adding a ground plane on the PCB of the pMUT array and use only shielded coaxial cables, however this only reduced the effect while keeping a significant capacitive signal in the water tank.

When implementing a communication scheme, both the ultrasonic signal and the capacitive signal will contribute to the communication link, and hence give us incorrect metric for both BER and SNR. For this reason, we implemented the QPSK channel only in the tissue phantom, where the capacitive signal is not present, and the ultrasonic pulse the main signal above the noise level. In fact, to verify that there is no capacitive coupling in the signal, we block the ultrasonic path with a solid block in the tissue phantom, and in that case no data can be transmitted, confirming that no capacitive effect is present. When repeating the same blocking experiment in the water tank, there is still that is transmitted on the communication link, hence it is not an appropriated medium to conduct this experiment.

Additionally, we believe that using a tissue phantom for the communication is better in terms of mimicking the impedance of the human tissue as well as the absorption/losses of the acoustic waves as in the response to the first comment. In fact, using the water as a medium will underestimate the losses, 0.002 dB/cm in water instead of 0.7 dB/cm for the average human tissue.

4. It will be useful to compare to AIN and LN in the context of bandwidth availability, and also to CMUTs, where the claim is often made of high bandwidth. Thinning down a thick LN wafer to 1 μ m maybe cost challenging.

We thank the reviewer for this question, and we apologize for the delay on responding. While our lab had AIN-based pMUT capabilities, we were waiting to set up a new sputtering tool in our cleanroom facility that allows us to dope the AIN with Scandium, and hence fabricated this new type of device for comparison. For this we added a new plot in the methods section as Figure 16 hereby shown. The impulse response is measured in a water tank as in Figure 13 and FFT is applied for bandwidth extraction at -6 dB from the peak as shown in Figure 2b after windowing out the capacitive response (for all pMUTs we verified that there is no such coupling in the tissue phantom). While LN-based pMUTs of this work show the highest bandwidth of 401 kHz, the 36% Sc-doped AIN shows very large bandwidth of 375 kHz as well, which makes it a good material to explore. However, a disadvantage of the Sc is the reproducibility of the sputtering process, which is still in its research infancy, while the advantage of the LN is that during the bonding process to the substrate, the LN maintains the same crystal structure and piezoelectric properties, hence making it more reliable and predictable.

Moreover, we agree with the reviewer that the Capacitive Micromachined Ultrasonic Transducers (cMUTs) can show a higher bandwidth than the pMUTs, even higher than 100%, thus with at a center frequency of 630 kHz, it makes its bandwidth larger than the bandwidth of the

LN pMUTs. However, in this work we choose to not compare the different type of pMUTs with the cMUTs because a priori we believe they are not a good fit for intrabody communication. The main reason being the biocompatibility since they require a large DC bias to function, and this pose concerns in terms of patient safety when implanted into the body. Additionally, they require higher power consumption than the pMUTs, hence requiring a larger implanted battery. Even though we believe the cMUTs are not suitable for intrabody application, we acknowledge their suitability for ultrasonic imaging when the transducer is not implanted, hence power and size is not a limiting factor.

Regarding the cost of the LN CMP to 1 μm , in our experience this is not more expensive than sputtering Sc-AlN or simple AlN. In fact, both steps are typical MEMS processes in a cleanroom facility, which scales down the cost when increasing the production volume, especially when scaling from a 4" wafer up to an 8" wafer. For this reason, we believe that fabricating LN-based pMUTs is not a limiting factor in terms of cost scaling.

Reviewer #2:

The authors present an ultrasound transducer made from lithium niobate that has resonant frequencies of ~400Khz. The transducer is used for ultrasonic intrabody communication.

1. The paper demonstrates a good application of lithium niobate material in a piezoelectric transducer configuration. It would be more suitable for an engineering journal e.g., IEEE as the focus is more on the systems development, implementation and testing of the transducer. The use of lithium niobate for ultrasonic transducers itself is not new, with several commercial products already available and patents dating more than a decade old.

We kindly thank the reviewer for this comment. While we agree that this work is suitable for IEEE journals as well, we believe that the development of Lithium Niobate (LN) based pMUTs brings a deeper innovation which makes it of interest to a broader audience. In fact, the focus of this work is that the piezoelectric crystal of the LN, when subject to a lateral electric field at this low frequency on a suspended membrane, it activates several vibration modes, which under a damped medium such as water or tissue phantom (hence human tissue as well), will merge into a large bandwidth of ~400 kHz. This bandwidth is larger than what we can achieve with more traditional Aluminum Nitride (AlN) based pMUTs as shown in the newly added Figure 16. Other materials look promising to achieve such bandwidth, such as Scandium-doped AlN at 36%, which the group has recently developed for comparison. However, we believe that the LN is more reliable during the fabrication process to maintain its piezoelectric properties, while the ScAlN is still in a research phase and different fabs and tools show different performances of the material. Nonetheless, AlN or ScAlN based pMUTs show only a pure resonance tone, hence there is no modes merging phenomenon happening when implanting the devices, which is what makes the LN-based pMUTs more innovative.

Furthermore, in Main section of the paper, we present a high-level description of the LN-based pMUTs and their application for intrabody communication, while keeping the more technical part in the Methods section, making this work more comprehensive to a broader audience. This is aligned with our vision that in the near future we will have more and more Implantable Medical Devices (IMDs) for continuous monitoring of patients which is a broad and interdisciplinary problem and cannot be solved just by very technical work, hence we would like to reach a broader audience that will help shape the future of this application.

Regarding the last part of the reviewer's comment, we agree and acknowledge the use of the LN material for other applications, such as RF resonators/filter, optical waveguides, etc. and including bulk ultrasonic transducers. However, there is no previous demonstration of miniaturized and micromachined ultrasonic transducers (pMUTs) based on X-cut LN such as in this work. The bulk LN transducers are more comparable to the already existing bulk PZT transducers, both posing limitations to their compatibility and implantation into a human body.

Reviewer #3 (Remarks to the Author):

1. The paper describes the design and construction of piezoelectric micromachined ultrasonic transducer arrays and experiments to explore the feasibility of high data rate ultrasonic communication based on these devices. The subject is of definite interest to the field and presents a promising approach for implanted devices requiring high data throughput. However, I would like to see clearer explanation of how this work advances that published in Herrera, B., Emrehan Demirors, G. Chen, R. Guida, Flavius V. Pop, Neil Dave, C. Cassella, T. Melodia and M. Rinaldi. "PMUT-BASED HIGH DATA RATE ULTRASONIC WIRELESS COMMUNICATION LINK FOR INTRA-BODY NETWORKS." (2018) and other related publications from the same team.

We kindly thank the reviewer for asking this question. The paper cited by the reviewer shows only some initial work done by our group regarding the implementation of ultrasonic communication link in a tissue phantom. That work was limited in bandwidth since it was demonstrated with bulk PZT ultrasonic transducers and with an Aluminum Nitride (AlN) pMUT. In this current work we improved the bandwidth significantly by completely changing the piezoelectric material from AlN to Lithium Niobate (LN) and bringing up a new generation of pMUTs that did not exist before. In particular, the focus of this work is that the piezoelectric crystal of the LN, when subject to a lateral electric field at this low frequency on a suspended membrane, it activates several vibration modes, which under a damped medium such as water or tissue phantom (hence human tissue as well), will merge into a large bandwidth of ~400 kHz. This bandwidth is larger than what we can be achieve with more traditional Aluminum Nitride (AlN) based pMUTs as shown in the newly added Figure 16.

Even though other materials look promising to achieve such bandwidth, such as Scandium-doped AlN at 36%, which the group has recently developed for comparison, we believe that the LN is more reliable during the fabrication process to maintain its piezoelectric properties, while the ScAlN is still in a research phase and different fabs and tools show different performances of the material. Nonetheless, AlN or ScAlN based pMUTs show only a pure resonance tone, hence there is no modes merging phenomenon happening when implanting the devices, which is what makes the LN-based pMUTs more innovative.

Moreover, in this work we improved the communication range as well compared to previously demonstrated pMUT intrabody link. This has been achieved by improving the receiving front-end circuitry, changing from a power amplifier topology to a charge amplifier followed by a voltage amplifier topology. Since the pMUTs are very high impedance devices ($\sim 6 \text{ k}\Omega$ @ 630 kHz, given a static capacitance of each pMUT around 200 fF, hence the array capacitance around 45 pF), the conversion sensitivity can be improved by using a low impedance transimpedance amplifier such as the charge-amp topology.

2. The work on the design and construction of PMUT devices is well described, reproducible and a valuable contribution. In water testing of frequency response and impulse response (using hydrophones) is convincing but I find it strange that no results are presented on the directivity of the device. I would expect to see measurements of the radiated beam pattern in at least one plane. This is clearly an important issue for practical application in implanted devices - how well aligned do the Tx and Rx arrays need to be to achieve the quoted performance?

We thank the reviewer for the great comment and question. The main reason why we did not perform a directivity measurement is because the 15x15 array shown in Figure 2d was designed with a pitch much smaller than the ultrasonic wavelength, making the array omnidirectional. For instance, given the speed of sound of 1500 m/s and the resonance frequency of 630 kHz, the wavelength results being 2.5 mm. The array of this work is 3x3 mm², where each pMUT is a 100x100 μm^2 suspended membrane, each separated with a pitch of 100 μm . This results in a pitch to wavelength ratio of 1/25, making the array mostly omnidirectional, as shown in the newly added Figure 1e. This figure shows the theoretical ultrasonic wave propagation in the XZ plane from the array (left figure) as well as the cross section in the X plane (right figure) at different distances from the pMUT array (1 to 13 cm). As we can notice from the cross section on the X plane, at 1 cm from the pMUT array, the ultrasonic waves have much higher intensity when aligned to the array, for example we lose around 12 dB when misaligned by 2 cm, and up to 19 dB when misaligned by 4 cm. The sensitivity to the misalignment goes down at longer distance and for example at 10 cm, we only lose 0.5 dB when misaligned by 2 cm and up to 1 dB when misaligned by 4 cm. In conclusion, given the extended implantation range that we cover in this experiment, we need to keep the transmitting and receiving transducers very well aligned otherwise the BER and SNR will degrade significantly, as shown in the newly added Figure 2d. To overcome this problem and minimize the misalignment when implanting a device in a real body, we have developed in another work* an intrabody discovery architecture, where an external transducer acts as a scanner to find the exact location of an implanted device. Once that device is found, the external transducer can be aligned and optimized the communication link.

*F. Pop, B. Herrera, C. Cassella and M. Rinaldi, "Enabling Real-Time Monitoring of Intrabody Networks Through the Acoustic Discovery Architecture," in IEEE Transactions on Ultrasonics, Ferroelectrics, and Frequency Control, vol. 67, no. 11, pp. 2336-2344, Nov. 2020, doi: 10.1109/TUFFC.2020.3002973.

Array Directivity - XZ plane and Cross Section

3. The communication performance is an area with some significant weaknesses. It would be more convincing if the channel impulse response and/or time domain signals for each range on the tissue phantom were presented (similar to their water tests) as well as the spectra. Not only does this help the reader to see if the communication channel is noise or multipath limited but appropriate propagation delay also confirms that the principal signal path is acoustic rather than electromagnetic crosstalk. (It is challenging to 100% eliminate EM coupling when using MHz frequency ultrasonic signals in an experiment with long cables, open PCB tracks and an unshielded tissue phantom). The spectrum of the signals remains remarkably flat over the range 3 - 13 cm when I would expect to see the differential attenuation between the low and high frequency band edges to be more visible, especially at the longest range. Again, seeing time domain signals would confirm that signal coupling was always through the expected mechanism.

We thank the reviewer for pointing this out. In order to respond to this comment, we acquired new data and compared the time domain response in a water tank experiment vs the tissue phantom. Here we drive the hydrophone with 2 sine wave cycles and received the signal on the LN pMUT array as shown in the figure below and in the manuscript's newly added Figure 17. As we can see from the top left plot, when in water, we have that the driving signal couples directly into the received signal through a capacitive coupling effect, while the received ultrasonic pulse is received with a delay of 20 μs , which indicated a distance of around 3 cm. In this same experiment, we blocked the ultrasonic path and in the top right figure we can see the ultrasonic pulse disappear while the capacitive coupling is still maintained. We tried to eliminate this issue in water by adding a ground plane to the pMUT array and using coaxial cables out of the water, but the coupling persisted. For this reason, we cannot conduct a communication experiment in a water tank since we the link will "see" the combine bandwidth of the capacitive and ultrasonic signal resulting in an incorrect BER vs SNR evaluation. However, in water measurements are still valid for bandwidth extraction since we can remove the coupling signal in post processing, which cannot be done during the communication since it's real-time data transmission.

At this point, when repeating the same experiments in a tissue phantom, the capacitive coupling is not present anymore. In the bottom left figure, we can see the ultrasonic pulse delayed by ~ 20 us at around 3 cm without any of the driving signal leaking into the received signal. Additionally, in the bottom right figure we are blocking once again the direct path in between the TX and RX and the ultrasonic pulse is strongly attenuated (there is still some small transmission since the blocking was done with a thick plastic object, which still allows for some ultrasonic transmission). Finally, we still kept the ground plane on the pMUT array PCB and the coaxial cables in order to avoid RF radiation. However, we believe that there is not much radiation at this low frequency, especially because the pMUT array only has two small wire bonds to the PCB with a very low inductance, and the PCM has only a small and shielded SMB connector. Given these results, we believe that our communication link in the tissue phantom is reliable in terms of only using the ultrasonic waves as a transmission channel and none of the capacitive coupling, hence the computed BER vs SNR are in good prediction. In fact, during the image transmission experiments, for sanity check, we block the transmission path with the same plastic block, and we can visually see how each pixel of the image disappears and becomes an error, making the image indistinguishable. Once the block is removed, the pixels start reappearing reducing the errors.

Regarding the second part of the comment about the signal attenuation at different frequencies, we agree with the reviewer that the bands don't have a large differential attenuation. The main reason is that the received signal at the array is proportional to only to the received pressure on each individual pMUT, but as well as to the operation frequency. Hence, even though higher ultrasonic frequencies attenuate more in the tissue phantom, at the same time we will have a slightly higher signal at higher frequencies, and thus compensating in part for the first attenuation. For this reason, we believe that the bands will not have a large difference between the left and right edge. However, if the bandwidth was to be much larger, spanning over several MHz, then the differential effect will be more visible.

4. Also, the quoted bit error rate of 10^{-1} at the maximum range is very high and, although appropriately designed error correction codes code makes this useable, this would be at the expense of a large reduction in data throughput since a quite low code rate would be required. Raw pixel data (allowing pixel errors) would be unlikely to be used in a practical imaging system and compressed images with error control coding are much more likely. These results need to be described in the context of delivering high integrity data (low error rate $< 10^{-5}$).

We thank the reviewer for this comment. The images that are transmitted are not envisioned to be used in an ultrasonic scanning system. The main reason why we chose an image and compute the pixel error, was to visually verify the quality of the channel in terms of BER, as well as making sure this was only ultrasonic data transmission. In fact, when blocking the ultrasonic path in a tissue phantom, all the received pixels are corrupt. However, when repeating the experiment in a water tank, where we know about the presence of the capacitive coupling, we still have many good, received pixels (low BER), hence this medium is not ideal for a communication link. We envision in this application to transmit encoded data streams and recover it through a Cyclic Redundancy Check (CRC) check method, which will do consume some portion of the data rate.

REVIEWERS' COMMENTS

Reviewer #3 (Remarks to the Author):

All comments have been taken on board and addressed in this revision, including new data gathered where necessary. Outstanding questions have been answered convincingly. I believe the article is now suitable for publication.

Reviewer #1

N/A

Reviewer #2

N/A

Reviewer #3

All comments have been taken on board and addressed in this revision, including new data gathered where necessary. Outstanding questions have been answered convincingly. I believe the article is now suitable for publication.

We thank the reviewer for the positive comment and for the suggestion to publish this article.